# Variational Autoencoders for Highly Multivariate Spatial Point Processes Intensities

**Baichuan Yuan**[1]**, Xiaowei Wang**[2]**, Jianxin Ma**[2]**, Chang Zhou**[2]**,**
**Andrea L. Bertozzi**[1]**, Hongxia Yang**[2]
[1]Department of Mathematics, University of California, Los Angeles
[2]DAMO Academy, Alibaba Group
`ybcmath@gmail.com, daemon.wxw@alibaba-inc.com,`
`majx13fromthu@gmail.com, ericzhou.zc@alibaba-inc.com,`
`bertozzi@math.ucla.edu, yang.yhx@alibaba-inc.com`

## Abstract

Multivariate spatial point process models can describe heterotopic data over space. However, highly multivariate intensities are computationally challenging due to the curse of dimensionality. To bridge this gap, we introduce a declustering based hidden variable model that leads to an efficient inference procedure via a variational autoencoder (VAE). We also prove that this model is a generalization of the VAE-based model for collaborative filtering. This leads to an interesting application of spatial point process models to recommender systems. Experimental results show the method's utility on both synthetic data and real-world data sets.

## 1 Introduction

Multivariate point processes are widely used to model events of multiple types occurring in an $n$ dimensional continuum. This paper focuses on multivariate spatial point processes (SPP), which can uncover hidden connections between subprocesses based on the correlations of their spatial point patterns. Often we encounter missing data problems, where some subprocesses are not fully observable. The underlying connections could further contribute to the prediction of these subprocesses over the unobserved areas. Moreno-Muñoz et al. (2018) has shown the effectiveness of this joint model for Gaussian processes with heterotopic data. Multi-output models in Lian et al. (2015) such as coregionalization and cokriging can outperform independent predictions. However, there is limited literature on the statistical methodology of the highly multivariate spatial point processes, according to the very recent paper (Choiruddin et al., 2019).

Inference for multivariate spatial point processes intensities is still a challenging problem (Taylor et al., 2015), especially with a large number of subprocesses. For popular Gaussian processes-based approaches (Williams & Rasmussen, 2006), the multivariate intensity often consists of independent and multi-output Gaussian processes. The complexity of the models and the curse of dimensionality hinder this approach for highly multivariate data, such as friendship networks and recommender systems with millions of users. In these problems, we only partially observe the events (e.g. users interact with items, locations) for each subprocess (user). It is necessary to jointly infer the preference of each user based on their hidden correlations. For example, a common approach in recommender systems, collaborative filtering (He et al., 2017), predicts the item interests of each user with the help of the collection of item preferences for a large number of users.

To address these problems, we propose a multivariate spatial point process model with a nonparametric intensity. We extend the well-known kernel estimator in Diggle (1985) to the multivariate case. This generalization is achieved through the introduction of hidden variables inspired by stochastic declustering (Zhuang et al., 2002). The latent variables naturally lead to a variational Bayesian inference approach, which is different from the frequentist point estimation in the kernel estimator. To reduce the complexity in the highly multivariate case, we consider an alternative set of hidden variables that are designed to work well as latent variables for a variational autoencoder (VAE) (Kingma & Welling, 2014). This amortized inference (Gershman & Goodman, 2014) approach leads to fast inference once the model is fully trained. Further, we show the equivalence for these two different

settings of hidden variables using the properties of the spatial point process. This efficient approach makes it possible to apply multivariate spatial point processes in many areas, including location-based social networks and recommender systems with many users. Moreover, the nonparametric method for analyzing spatial point data patterns is not related to specific parametric families of models, which only requires the intensity to be well-defined.

Our approach is not a direct replacement for current inference methods on few-variate spatial point processes (Jalilian et al., 2015). In contrast to the classical methodology, VAE requires a large number of training data. The highly multivariate data that are widely available in social networks and recommender systems can be ideal applications for our approach. In fact, it can be shown that our model is a generalization of a state-of-the-art VAE-based collaborative filtering model (Liang et al., 2018). Our model nonparametrically fits the underlying intensity function. Compared with the multinomial distribution used in Liang et al. (2018), this leads to not only a smoother intensity over space but also better predictions in terms of ranking-based losses. Compared to a univariate model, such as the trans-Gaussian Cox processes (Williams & Rasmussen, 2006), our multivariate model enhances the predictive ability on missing or unobserved areas, which is consistent with the results of heterogeneous multi-output Gaussian processes (Moreno-Muñoz et al., 2018).

The contributions of this paper are three-fold. We first build a novel multivariate spatial point process model and find a direct connection with the VAE-based collaborative filtering through detailed theoretical analysis. Secondly, this connection introduces amortized inference for an efficient multivariate point process estimation. Finally, point processes generalize the discrete distribution used in (Liang et al., 2018) and lead to a better modeling of spatial heterogeneity. We validate these benefits through experiments for multiple multivariate data sets, showing improvement over classic SPP methods and potentials on collaborative filtering applications.

## 2 PRELIMINARIES

**Spatial point process**  A *point process* (PP) is a random counting measure $N(x)$ on a complete separable metric space $R$ (here we always assume that $R \subset \mathbb{R}^n$) that takes values on $\{0, 1, 2, ...\} \bigcup \{\infty\}$. While the major theory of point processes centers around the temporal dynamics, spatial point process models (Diggle et al., 1983) are established in forestry and seismology, focusing on the stationary and isotropic case. We focus on the (first-order) *intensity function* $\lambda(x)$, which is the expected rate of the accumulation of points around a particular spatial location $x$. We write

$$\lambda(x) = \lim_{|\Delta x| \downarrow 0} \frac{\mathbb{E}\left[N(\Delta x)\right]}{|\Delta x|}, \tag{1}$$

where $\Delta x$ is a small ball in the metric space, e.g. the Euclidean space $\mathbb{R}^n$, with the centre $x$ and measure $|\Delta x|$. The second-order intensity function is naturally defined as

$$\lambda_{(2)}(x, y) = \lim_{|\Delta x|, |\Delta y| \downarrow 0} \frac{\mathbb{E}\left[N(\Delta x)N(\Delta y)\right]}{|\Delta x||\Delta y|}, \tag{2}$$

measuring the chance of points co-occurring in both $\Delta x$ and $\Delta y$. Normalizing this leads to the *pair-correlation function* $g(x, y) = \lambda_{(2)}(x, y)/\lambda(x)\lambda(y)$. $g(x, y) > 1$ indicates that points are more likely to form clusters than the simple Poisson process where $g(x, y) = 1$.

Common models in SPPs include the Poisson process with a non-stationary rate $\lambda(x)$, and the Cox process with a nonnegative-valued *intensity process* $\Lambda(x)$, which is also a stochastic process. Cox processes conditional on a realization of the intensity process $\Lambda(x) = \lambda(x)$ are Poisson processes with intensity $\lambda(x)$. To model the aggregated points patterns, Poisson cluster (Neyman-Scott) processes generate parent events from a Poisson process. Then each parent independently generates a random number of offsprings. The relative positions of these offsprings to the parent are distributed according to some p.d.f $K_\sigma(x)$ in space (Diggle et al., 1983). Many point process models, including most Cox processes, are in fact Poisson cluster processes. The duality between Cox processes and cluster processes is widely used to construct Cox process models. For example, the kernel-based intensity process $\Lambda(x) = \sum_{i=1}^{\infty} K_\sigma(x - x_i)$ with $x_i$ from a Poisson process, is essentially a Poisson cluster process. The number of offsprings is from a Poisson distribution with $\lambda = 1$ and the relative position distribution is $K_\sigma(x)$. Repulsive SPPs, on the other hand, model that nearby points of the

process tend to repel each other. Higher order intensities are often considered in this case, such as determinantal PPs.

Alternatively, if we are more interested in the realization intensity $\lambda(x)$ than the mechanical interpretation, the trans-Gaussian Cox process provides a tractable way to construct the Cox process using a nonlinear transformation on a Gaussian process $S(x)$. Popular choices for $\Lambda(x)$ include the log-Gaussian Cox process (LGCP) with $\exp(S(x))$ and the permanental process with $S(x)^2$. Recent works on Cox processes have been extensively focused on the cases that are modulated via the Gaussian random field, due to its capability in modeling the intensity and pair-correlations between subprocesses. In the next section, we develop a more explicit approach to model interactions for fast inference and the generalization ability for new subprocesses.

**Inference for Point Processes**  Inference methods for point processes are mainly based on the order statistics or likelihood function. The order statistics are often estimated nonparametrically, such as the kernel estimator (Diggle, 1985) of the intensity function. For the likelihood-based inference, we assume that one observes events $X = \{x_i\}_{i=1}^{N}$ of the underlying spatial point process over the area $R$. The log-likelihood for the inhomogeneous Poisson process over space $R$ is

$$\log p(X|\Theta) = \sum_{i=1}^{N} \log(\lambda(x_i)) - \int_R \lambda(x)\, \mathrm{d}x\,. \tag{3}$$

The integration term is the log void probability and can be viewed as a normalization term for the likelihood. For Cox processes, the likelihood is the expectation over the Poisson likelihood. It is difficult to directly integrate over the distribution of $\Lambda$. Monte Carlo methods (Adams et al., 2009) are commonly used to approximate the expectation. To improve the scalability of the expensive sampling, many methods such as variational inference (Lloyd et al., 2015), Laplace approximation (Williams & Rasmussen, 2006) and reproducing kernel Hilbert spaces (Flaxman et al., 2017) are proposed.

**Variational Autoencoder**  As a stochastic variational inference algorithm, VAE (Kingma & Welling, 2014) is maximizing the evidence lower bound (ELBO) of the log-likelihood function

$$\log p(X|\Theta) \geq \mathbb{E}_{q_\phi(z|X)}[log(p_\theta(X|z)] - KL(q_\phi(z|X)|p(z)). \tag{4}$$

The hidden variables $z$ have a simple multivariate Gaussian prior $p(z) = \mathcal{N}(z; 0, I)$. The true posterior, which is often intractable as in the Cox process, is approximated via a multivariate Gaussian $q_\phi(z|X) = \mathcal{N}(z; \mu_\phi(X), \sigma_\phi(X))$. The KL divergence term in the ELBO can be calculated analytically. VAE uses a multilayer perceptron (MLP) to learn the mean and variance of the approximated posterior directly from the data. The most related work here is a recent VAE-based model for collaborative filtering (VAE-CF) (Liang et al., 2018). They assume that each user is a multinomial distribution over items with the log-likelihood $\log p_\theta(X_u|z_u) = \sum_{i=1}^{N} X_{iu} \log \pi_i(z_u)$ for each user $u$. Here $X_u$ is the observed data of user clicking items, $\pi_i(z_u)$ is the probability that user $u$ clicks the item $i$ and $X_{iu}$ is an indicator function on whether the user $u$ clicked the item $i$.

## 3 MULTIVARIATE SPATIAL POINT PROCESSES

Here we consider a multivariate case of the SPP, with $U$ interdependent univariate point processes on the sample space $R$. The intensity function is measured in a similar way as the univariate case via $\lambda_u(x) = \lim_{|\Delta x| \downarrow 0} \left( \mathbb{E}\left[ N_u(\Delta x) \right] / |\Delta x| \right)$, where $N_u(\Delta x)$ is the number of events within a set $\Delta x$ for the subprocess $u$.

### 3.1 A NONPARAMETRIC MODEL

The observed data of multivariate SPP include the location of $N_u$ events $X_u = \{x_i^u\}_{i=1}^{N_u}$ associated with each subprocess $u$. For each $u$, the observed event locations follow a Poisson process with spatial intensity $\lambda_u(x)$, which is a realization of the random intensity $\Lambda_u(x)$. Using the nonparameteric kernel estimator, the intensity of the subprocess $u$ is estimated by

$$\lambda_u(x) = \sum_{i=1}^{N_u} K_\sigma(x - x_i^u). \tag{5}$$

Here $K_\sigma(x)$ is a kernel function and we usually adopt the radial basis function kernel (RBF) where $K_\sigma(x) = \exp(-\|x\|^2/2\sigma^2)$. We ignore the end-correction (Diggle, 1985) for now.

In real-world applications, however, one often encounters the missing data problem, where we cannot directly observe points in certain areas for some subprocesses. Instead, we seek to infer the hidden data from other fully observed subprocesses. In our model, we assume that each subprocess reflects the stochastic and heterogeneous patterns. For example, users in an e-commerce platform usually prefer different categories. As in Poisson cluster processes, this naturally leads to events clustering in specific areas. Another real-world example is the aggregation of check-in activities around the home and workplaces for social network users (Cho et al., 2011). Note that $N = \sum_{u=1}^{U} N_u$ is the total number of events. We introduce hidden variables $Y_i^u$ for each event $x_i = 1, ..., N$ and subprocess $u = 1, ..., U$, where $Y_i^u = 1$ if the subprocess $u$ includes event $x_i$ and $Y_i^u = 0$ otherwise. $\mathbb{E}Y_i^u = p_i^u$ is the probability that event $x_i$ is from the subprocess $u$. Then the intensity process for our multivariate SPP model is

$$\Lambda_u(x) = \sum_{i=1}^{N} Y_i^u K_\sigma(x - x_i), \tag{6}$$

for each subprocess $u$. This model generalizes the kernel density-based intensity to the missing data case. Similarly to the original method, it can be applied to estimate the intensity for both cluster processes such as Cox processes and repulsive ones like determinantal PPs. In order to incorporate prior information and model the data uncertainty, we adopt a variational inference approach for the hidden variables.

## 3.2 VARIATIONAL INFERENCE

A major drawback of current inference methods for SPP is the introduction of a large number of parameters in the highly multivariate case. For our model, we use an amortized inference approach - VAE (Kingma & Welling, 2014) to avoid the computational complexity of directly estimating the posterior for each subprocess $u$.

The generative process of our model can be described as follows: For each subprocess $u$, it has a $K$-dimensional hidden variable $z_u$ with a multivariate normal prior $z_u \sim \mathcal{N}(0, I_K)$. Here we use a low-dimensional representation and then a nonlinear mapping $f_\theta(z_u) = \{p_i^u\}_{i=1}^{N}$ transforms $z_u$ so that it has the same dimension as the number of events $N$. Finally the spatial points of the subprocess $u$ are sampled according to the intensity $\lambda_u(x) = \sum_{i=1}^{N} p_i^u K_\sigma(x - x_i)$. We approximate the intractable posterior distribution of $z$, $q(z|X)$ with a multivariate Gaussian $\mathcal{N}(\mu_\phi(X), \sigma_\phi(X))$. As in Liang et al. (2018), we use MLPs to learn the nonlinear function $f_\theta(z)$ with parameters $\theta$ and the mean and variance with parameters $\phi$. The variational bound of our multivariate Cox process model is then

$$\log p(X_u|\Theta) \geq \mathbb{E}_{q_\phi(z_u|X_u)}[log(p_\theta(X_u|z_u)] - KL(q_\phi(z_u|X_u)|p(z_u)) = \mathbf{L}. \tag{7}$$

The first term in $\mathbf{L}$ is essentially a complete likelihood function. For each subprocess $u$, it has the following (expected) intensity function

$$\mathbb{E}_{q_\phi(z_u|X_u)}\Lambda_u(x) = \sum_{i=1}^{N_u} p_i^u K_\sigma(x - x_i^u) \tag{8}$$

and a Poisson process log-likelihood function from (8) and (3)

$$\mathbb{E}_{q_\phi(z_u|X_u)} \log p_\theta(X_u|z_u) = \sum_{i=1}^{N_u} \log(\sum_{i=1}^{N_u} p_i^u K_\sigma(x - x_i^u)) - \int_R \sum_{i=1}^{N_u} p_i^u K_\sigma(x - x_i^u)dx. \tag{9}$$

For applications without explicit spatial information, we embed each event into a latent space as a vector. First, we obtain a similarity graph for all events. Then the embedding $x_i$ of $i_{th}$ event in this graph is obtained via graph neural networks (GNNs) such as GraphSAGE (Hamilton et al., 2017). See Figure 1 for an illustration of our framework. Both $x_i$ and $z_u$ are learned jointly combining the information of item embedding and user hidden variable.

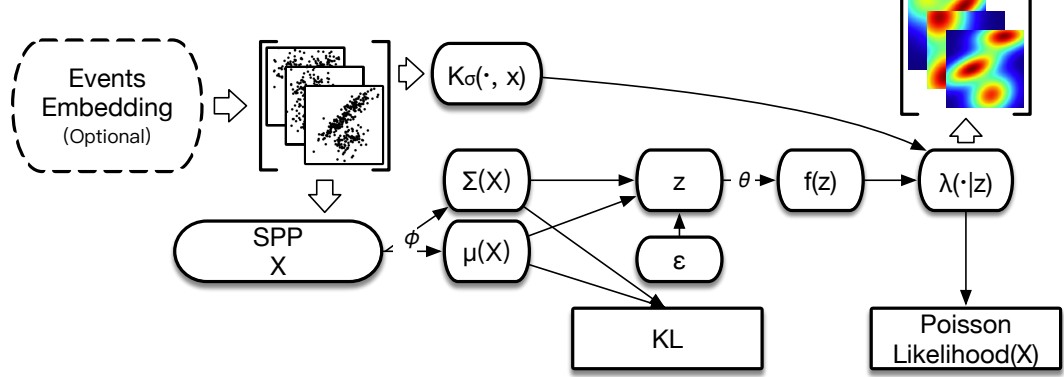

Figure 1: Visual illustration of our spatial point process model via VAE during the training.

## 3.3 ALTERNATIVE MODEL

Recall that the hidden variables $Y_i^u$ describe whether the event $x_i$ is from the subprocess $u$. By definition, we have $\sum_{u=1}^{U} Y_i^u = 1$ and $\sum_{u=1}^{U} p_i^u = 1$ for any $i$. During the training process, it is difficult to normalize the probability $p_i^u$ over all subprocesses (have to use the full data). Moreover, this constraint leads to $g_{uv} < 1$ for $u \neq v$, implying mutual-inhibition behaviors between subprocesses. Instead, we consider an alternative model where $p_i^u$ is the probability that the subprocess $u$ generates an event $x_i$ i.e. $\sum_{i=1}^{N_u} p_i^u = 1$ for each $u$. During the training, the total number of events $N_u$ is not viewed as a hidden variable for each subprocess. Thus the alternative model essentially normalizes $\lambda_u$ by a constant. With the reparameterization trick in Kingma & Welling (2014), we sample the log-likelihood function using all events within a mini user batch and compute the gradient. This approach incorporates all information about the user so that negative sampling is not needed. See Algorithm 1 for our training procedure. For the model prediction, the normalized intensity of a new subprocess can be efficiently calculated in $O(N)$ using the approximated posterior $q_\phi(z|X_{new})$ and nonlinear function $f_\theta(z)$ with parameters $\theta, \phi$ inferred from data. We can further reduce the computational challenge (Liang et al., 2018) for large $N$ due to $f_\theta(z)$ by discretizing the space.

Now we show the equivalence of our multivariate model and the alternative one. There are two probabilities to consider. The first one is the conditional probability of observed events $X_u$ in the subprocess $u$ with an intensity function $\lambda_u(x)$, given that there are $N_u$ events within the metric space $R$. The second one is the probability of sample $X_u$ of size $N_u$ from the normalized density $h_u(x) = \lambda_u(x)/\int_R \lambda_u(s)ds$. For general SPPs data, we have

**Theorem 1.** *A spatial point process on a measurable set $R \subset \mathbb{R}^n$ with an intensity function $\lambda_u(x)$ is equivalent to $N_u$ i.i.d samples within $R$ with p.d.f $h_u(x) = \lambda_u(x)/\int_R \lambda_u(s)ds$, given we know $N_u = \int \lambda_u(s)ds$, which is the number of points within $R$ for the point process model.*

*Proof.* See Section B in Appendix. □

According to this theorem (see supplementary material), we can replace the log-likelihood function (9) in the ELBO with

$$\mathbb{E}_{q_\phi(z_u|X_u)} \log p_\theta(X_u|z_u) = \sum_{i=1}^{N_u} \log(h_u(x_i^u)) + C. \qquad (10)$$

Here $C$ is related to the log-likelihood on the number of events $N_u$, which is a constant because $\int_R \lambda_u(s)ds = N_u$ is observed. One drawback of this approach is that, for the prediction of actual missing data, we cannot infer the number of missing points. Instead, our VAE-based model generates the normalized intensity predicting the possible locations for the missing events. We use the alternative definition of $p_i^u$ from now on.

This result shows that VAE-CF is a special case of this multivariate SPP model over a discrete space $X$ of events. In fact, VAE-CF is the alternative model with a delta function as the kernel $(h_u(x_i) = \lambda_u(x_i)/\int_X \lambda_u(s)ds = p_i^u)$, which is equivalent to the SPP model according to the theorem. To better model the spatial heterogeneity of events, one can replace the delta function with other kernels or use more advanced SPP intensities. We simply use a RBF kernel here, resulting in $h_u(x) = \sum_{i=1}^N p_i^u \exp(\|x - x_i\|^2/2\sigma^2)$.

---

**Algorithm 1:** Training VAE SPP with stochastic gradient descent.

---

**Input:** Traning subprocesses $u \in \mathbf{U_T}$ with their point locations $X_u$
**Result:** Parameters $\theta$ and $\phi$
Initialize $\theta$ and $\phi$ randomly;
**while** *not converged* **do**
    Sample a subprocesses batch $\mathbf{U_b}$ from $\mathbf{U_T}$ and their points $X_b = \bigcup_{u \in \mathbf{U_b}} X_u$ ;
    **forall** $u \in \mathbf{U_b}$ **do**
        Sample $z_u \sim \mathcal{N}(\mu_\phi(X_u), \sigma_\phi(X_u))$ with reparameterization trick;
        Compute $f_\theta(z_u) = \{p_i^u\}_{x_i \in X_b}$;
        **forall** $x \in X_u$ **do**
            Compute sampled normalized intensity $h_u(x) \approx \sum_{x_i \in X_b} p_i^u K_\sigma(x - x_i)$;
        **end**
        Compute noisy gradients of the ELBO $\mathbf{L}$ w.r.t $\theta$ and $\phi$
    **end**
    Average noisy gradients over batch;
    Update $\theta$ and $\phi$ with the Adam optimizer (Kingma & Ba, 2015);
**end**

---

One benefit of this alternative model is its resulted consistency. The nonparametric kernel estimation for the point process intensity is unbiased. To see this, for any measurable set $R$, we take the expectation of the estimated intensity $\lambda(x)$ over the Poisson point process distribution

$$\mathbb{E} \int_R \lambda(x)dx = \int_R \mathbb{E} \sum_{i=1}^N K_\sigma(x - x_i)dx = \int_R \int_R K_\sigma(x - y)\rho(y)dydx = \int_R \rho(y)dy, \quad (11)$$

where $\rho(y)$ is the true intensity function. Then $\mathbb{E}\lambda(x) = \rho(x)$ under mild conditions, e.g., a spatially continuous assumption on $\rho$. But it is inconsistent due to the non-vanishing variance without normalization. For our alternative model, the normalized intensity function $h_u(x)$ is still unbiased. And according to the standard theory of the multivariate kernel density estimation (KDE), the consistency of $h_u(x)$ is also guaranteed. Another benefit of using this alternative form can be seen from the cross pair-correlation function. For the alternative model, we remove the undesirable restriction of negative correlations between all users ($g_{uv} < 1$ for $u \neq v$) and can incorporate more diverse relationships between users. To see this, we first consider the *auto* and *cross pair-correlation function* $g_{uv} = \mathbb{E}\Lambda_u\Lambda_v/\mathbb{E}\Lambda_u\mathbb{E}\Lambda_v$. For our original model, it is straightforward to prove that $g_{uu} > 1$ and $g_{uv} < 1, u \neq v$ (see supplementary material). The auto pair-correlation functions show that our model is more aggregate than the simple Poisson process.

## 4 EXPERIMENTS

We compare our model (with the RBF kernel, VAE-SPP) with both VAE-CF (Liang et al., 2018) and univariate spatial point process models using a standard KDE (Diggle, 1985) or TGCP (Williams & Rasmussen, 2006) as intensity functions. We adopt the experiment setting in VAE-CF. We split the data into training, validation and testing sets. For the multivariate model, the training data is used to learn the parameters $\theta, \phi$. For KDE and TGCP models, we omit the training data because different subprocesses are assumed to be independent and also because of the computational complexity of fitting a highly multivariate TGCP. We assume that only $80\%$ of the events in the validation and test sets are observed. The remaining $20\%$ are viewed as missing data to be inferred by different models. Hyperparameters are selected on the validation data as in Liang et al. (2018). Finally, we compare the prediction performance of different models on the missing data given the partially-observed events. We use standard ranking losses such as NDCG@K and Recall@K defined in Appendix D.1.

## 4.1 MULTIVARIATE SPP ON SPATIAL DATA

**Synthetic data sets** We simulate two different data sets using multiexponential and multisine models. For the multiexponential data set, we simulate 5,000 Poisson processes with $\lambda_k(x) = a_k e^{-b_k x}, k = 1, ..., 5000, x \in [0, 30]$ as training data. Here $a_k$ and $b_k$ are uniformly sampled between $[5, 10]$ and $[0.1, 0.2]$ separately. 500 validation and 500 test subprocesses are generated in the same way with parameters sampled from $a_k$ and $b_k$. The multisine data set is generated via replacing the intensity function with $\lambda_k(x) = \max(a_k * \sin(b_k x), 5)$ and sampling $a_k$ and $b_k$ uniformly between $[5, 10]$ and $[1, 2]$ separately. Each realization of the spatial point process is discretized using a uniform grid over $x$ with grid spacing 0.01.

Table 1: Testing results on the simulation data sets. Both the mean and variance are percentages (same below).

| | Multiexp | | | Multisine | | |
|---|---|---|---|---|---|---|
| Name | NDCG@100 | Recall@50 | Recall@100 | NDCG@100 | Recall@50 | Recall@100 |
| VAE-CF | 6.78(0.28) | 7.25(0.40) | 14.5(0.52) | 3.30(0.15) | 2.49(0.13) | 4.64(0.18) |
| VAE-SPP | **7.11**(0.31) | **7.34**(0.40) | **14.9**(0.54) | 3.53(0.15) | **2.58**(0.13) | **4.90**(0.18) |
| KDE | 5.27(0.15) | 5.85(0.12) | 11.8(0.17) | 3.23(0.15) | 2.29(0.12) | 4.55(0.27) |
| TCGP | 3.11(0.14) | 3.32(0.11) | 6.44(0.11) | **3.77**(0.14) | 1.88(0.11) | 3.92(0.17) |

**Location-based Social Network.** We consider the Gowalla data set (Cho et al., 2011) in New York City (NYC) and California (CA). We use a bounding box of -124.4096, 32.5343, -114.1308, 42.0095 for CA and -74.0479, 40.6829, -73.9067, 40.8820 for NYC (both from flickr[1]). Each user with at least 20 events (check-ins) is viewed as a subprocess. There are 673,183 events and 6,728 users for Gowalla-CA. We randomly select 500 users as the validation set and 500 users as the testing set. We use the remaining users for training. For Gowalla NYC, there are 86,703 events from 1,171 users. We set the size of both validation and testing sets to 100. For the spatial tessellation, we use uniform grids ($32 \times 32$ for NYC and $64 \times 64$ for CA). Both our model and VAE-CF can work without grids. We further compare the performance of our model with VAE-CF by viewing each location as an item.

Table 2: Testing results on the Gowalla data sets with uniform grids.

| | CA | | | NYC | | |
|---|---|---|---|---|---|---|
| Name | NDCG@100 | Recall@50 | Recall@100 | NDCG@100 | Recall@50 | Recall@100 |
| VAE-CF | 41.8(1.5) | 64.8(2.0) | 70.0(2.0) | 43.6(2.3) | 73.9(2.9) | **86.2**(2.2) |
| VAE-SPP | **42.3**(1.5) | **65.2**(2.0) | **70.2**(1.9) | **44.8**(2.4) | **74.5**(2.9) | **86.2**(2.2) |
| KDE | 34.5(1.5) | 59.2(2.0) | 64.0(2.0) | 41.2(1.5) | 69.9(2.0) | 83.6(2.0) |
| TCGP | 31.8(1.3) | 56.5(2.0) | 60.9(2.0) | 37.3(2.3) | 59.9(3.3) | 75.9(2.8) |

Table 3: Testing results on the Gowalla data sets without discretization.

| | CA | | | NYC | | |
|---|---|---|---|---|---|---|
| Name | NDCG@100 | Recall@20 | Recall@100 | NDCG@100 | Recall@20 | Recall@100 |
| VAE-CF | 21.3(0.77) | 16.6(0.74) | 32.8(0.97) | 16.0(1.7) | 13.2(1.7) | 26.3(2.4) |
| VAE-SPP | **21.6**(0.77) | **17.0**(0.80) | **33.5**(0.76) | **16.1**(1.7) | **13.7**(1.8) | **27.1**(2.5) |

In Table 1, we summarize the performance of both multivariate and univariate models on the simulation data sets. It is clear that the multivariate models outperform the univariate ones. Moreover, testing on multivariate models takes less time because it only evaluates the posterior probability and intensity function. This illustrates the power of multivariate models using amortized inference. Within the multivariate models, our continuous model further improves upon the discrete VAE-CF. This is due to the fact that these simulation intensities are continuous over $R$. For real-world applications, the results on the location-based social network prediction and recommendation with and without grids are presented in Table 2 and 3. We observe the same pattern in both NYC and CA.

---

[1]https://www.flickr.com/places/info/

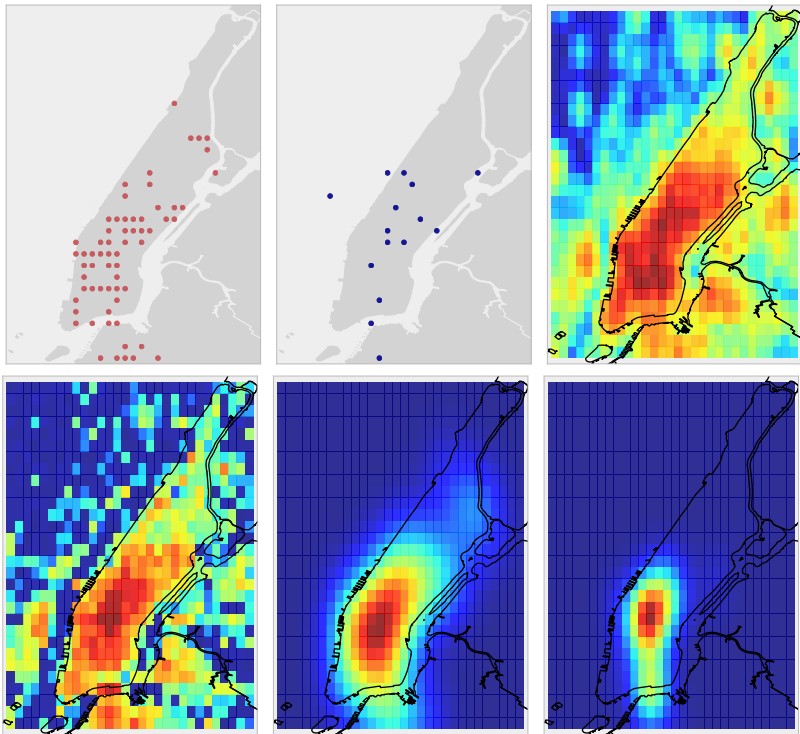

Figure 2: Estimated density functions for a Gowalla user in NYC (log scale). The first row from left to right: observed check-in locations (in red), held-out check-in locations (in blue, as missing data) and the estimated intensity from VAE-SPP. The second row from left to right: the estimated intensity (or density) from VAE-CF, KDE and TGCP.

We stop using univariate models from now on due to their inferior performances, especially for collaborative filtering applications. Moreover, our model improves discrete VAE-CF regardless of the choice of spatial grids. For visualization purposes, in Figure 2, we plot a user's check-in locations in Gowalla-NYC and intensities estimated via different methods. Comparing with VAE-CF, our model generates a continuous intensity. The univariate models overfit the training data and lead to inferior predictions of the missing data.

## 4.2 MULTIVARIATE SPP WITH A LATENT SPACE

MovieLens data sets (ML-100K and ML-1M) include the movie (item) rating by users and we binarize the rating with a threshold of 4. In the spatial point process setting, we view each user as a subprocess over the latent space of item embeddings. Here the item embedding is generated via a GNN. This framework is a natural generalization of the multimodal distribution over items. The item-item graph is constructed based on item-item similarities. We use the Jaccard distance to measure the similarities between items, which are further viewed as the sampling probabilities for GNN. Currently, we only consider 1-hop connections. Both GNN and VAE are trained jointly, which is more expensive than VAE-CF but leads to better performance compared to separate training (see Appendix D). For movie recommendation tasks, we compare the discrete VAE-CF to our joint model with GNN. The results in Table 4 show again the improvement of our model over the baseline.

Table 4: Testing results on the MovieLens data sets.

| Name | ML100K | | | ML1M | | |
|---|---|---|---|---|---|---|
| | NDCG@100 | Recall@20 | Recall@100 | NDCG@100 | Recall@20 | Recall@100 |
| VAE-CF | 40.8(2.8) | **32.3**(2.8) | 57.6(3.3) | 41.6(0.76) | 33.1(0.81) | 56.8(0.88) |
| VAE-SPP | **41.5**(2.9) | 31.3(2.7) | **59.0**(3.5) | **42.3**(0.77) | **33.9**(0.82) | **57.6**(0.88) |

## 5 CONCLUSION

In this paper, we introduce a novel spatial point process model for efficient inference on the highly multivariate case. Through amortized inference, our model makes it possible to investigate correlations between a myriad of point patterns based on a large number of training data, and the theoretical analysis on the density and intensity for SPPs builds the connection between our model and VAE-CF. There are many promising directions of future works including the extension for multivariate spatiotemporal PPs (Mohler et al., 2011; Yuan et al., 2019) and using features as covariances. There are multiple ways to estimate the mean rate ($h_u(x)$) of a spatial point process overall events, including Gaussian mixture models, Gaussian processes and flow-based models. For future work, we can investigate the connections between our model and other density-based estimations for point processes. Another interesting application is to handle real-world recommender systems via improving the joint training efficiency and comparing thoroughly with simpler algorithms as in Dacrema et al. (2019).

ACKNOWLEDGMENTS

We would like to thank the comments from Frederic P. Schoenberg and George Mohler. Andrea L. Bertozzi and Baichuan Yuan want to thank the support of NIJ fellowship 2018-R2-CX-0013 and NSF DMS-1737770.

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

# A  TABLE OF NOTATIONS

Table 5: Notations.

| Notation | Definition or Descriptions |
|----------|----------------------------|
| $N(x)$ | counting measure on a metric space $R$ |
| $N_u$ | the number of events of subprocess $u$ |
| $\lambda_u(x)$ | intensity function of a subprocess $u$ |
| $\Lambda_u(x)$ | intensity process of a subprocess $u$ |
| $K_\sigma(x)$ | kernel function |
| $U$ | # of subprocesses on the space |
| $\mathbf{U}_b$ | subprocesses in a batch |
| $X$ | events set |
| $X_u$ | observed events of subprocess $u$ |
| $X_b$ | all the observed events in a batch of subprocesses |
| $x_i$ | embedding/location of the $i_{th}$ event |
| $Y_i^u$ | hidden variables indicate whether the subprocess $u$ includes the $i_{th}$ event |
| $z_u$ | K-dimensional hidden variable represents subprocess $u$ |
| $p_i^u$ | probability of the $i_{th}$ event occurs in subprocess $u$ |
| $\phi, \theta$ | parameters of encoder($\mu_\phi, \sigma_\phi$) and decoder($f_\theta$) |
| $g_{uv} = \mathbb{E}\Lambda_u\Lambda_v/\mathbb{E}\Lambda_u\mathbb{E}\Lambda_v$ | auto and cross pair-correlation function |
| $h_u(x)$ | normalized density |

# B  PROOF OF THEOREM 1

*Proof.* We define our model as a point process on R with the intensity function $\lambda_u(x)$.

The alternative model is $N_u$ i.i.d samples within R with p.d.f $h_u(x)$, given that we know $N_u = \int \lambda_u(s)ds$ is the number of points within the point process model.

1) Our model has the following probability generating functional

$$G(v) = \exp(-\int_{\mathbf{R}^d}[1 - v(x)]\,\Lambda(\mathrm{d}x)) \tag{12}$$

2) Given $N_u$,

$$p(x_1, ..., x_{N_u}|N_u) = \prod_{i=1}^{N_u} h_u(x_i) \tag{13}$$

3) Our alternative model (a counting r.v. $N(x)$ with locations according to $h_u(x)$) has the following characteristic functional

$$G_c(v) = \sum_{n=0}^{\infty} p(N(R) = n)\mathbb{E}[\exp(\int_R \log(v(s))N(ds))|N(R) = n] \tag{14}$$

Using 2), we can evaluate this conditional probability

$$\mathbb{E}[\exp(\int_R \log(v(s))N(ds))|N(R) = n] = (\frac{\int_R \lambda_u(s)v(s)ds}{\lambda_u(s)ds})^n \tag{15}$$

Using 2) again and because the point process observation probability is

$$p(\omega) = p(N(R) = N_u)p(x_1, ..., x_{N_u}|N_u) = \frac{1}{n!}[\prod_{i=1}^{n} \lambda_u(x_i)] \exp(-\int_R \lambda(x)dx), \tag{16}$$

we have

$$G_c(v) = \exp(-\int_R \lambda(x)dx)(1 + \sum_{n=1}^{\infty} \frac{1}{n!}(\int_R \lambda(s)v(s)ds)^n) = \exp(\int_R \lambda(s)(v(s) - 1)ds). \tag{17}$$

The theorem follows from $G_c(v) = G(v)$ as the probability generating functional completely determines the probability structure of the point process. $\square$

We show that (10) holds in the main paper.

**Corollary 1.1.**

$$\mathbb{E}_{q_\phi(z_u|X_u)} \log p(X_u|z_u) = \sum_{i=1}^{N_u} \log(h_u(x_i^u)) + C. \tag{18}$$

*Proof.* Define $\lambda_u(x) = \mathbb{E}_{q_\phi(z_u|X_u)}\Lambda_u(x)$.

$$\mathbb{E}_{q_\phi(z_u|X_u)} \log p(X_u|z_u) = \log\left(p(N(R) = N_u)p(x_1^u, ..., x_{N_u}^u|N_u)\right) \tag{19}$$

$$= \sum_{i=1}^{N_u} \log(h_u(x_i^u)) + \log(p(N(R) = N_u)) \tag{20}$$

$$\log(p(N(R) = N_u)) = n \log(\int_R \lambda(x)dx) - \log(n!) - \int_R \lambda(x)dx \tag{21}$$

is only a function of $N_u$. $\square$

## C   AUTO AND CROSS PAIR-CORRELATION FUNCTIONS

We show that $g_{u,v}(x, y) > 1$ for $u = v$ and $g_{u,v}(x, y) < 1$ for $u \neq v$ for our orginal model.

$$g_{u,v}(x, y) = \frac{\mathbb{E}\Lambda_u(x)\Lambda_v(x)}{\mathbb{E}\Lambda_u(x)\mathbb{E}\Lambda_v(x)} \tag{22}$$

We have

$$\mathbb{E}\Lambda_u(x)\Lambda_v(x) = \mathbb{E}(\sum_{i=1}^{N} Y_i^u K_h(x - x_i))(\sum_{j=1}^{N} Y_j^v K_h(y - x_j)) \tag{23}$$

$$= \sum_{i=1}^{N}\sum_{j=1}^{N} \mathbb{E}Y_i^u Y_j^v K_h(x - x_i)K_h(y - x_j). \tag{24}$$

Similarly,

$$\mathbb{E}\Lambda_u(x)\mathbb{E}\Lambda_v(x) = (\mathbb{E}\sum_{i=1}^{N} Y_i^u K_h(x - x_i))(\mathbb{E}\sum_{j=1}^{N} Y_j^v K_h(y - x_j)) \tag{25}$$

$$= \sum_{i=1}^{N}\sum_{j=1}^{N} p_i^u p_j^v K_h(x - x_i)K_h(y - x_j). \tag{26}$$

Note that $\sum_{u=1}^{U} Y_i^u = 1$ and $\sum_{u=1}^{U} p_i^u = 1$. When $i \neq j$, we have $\mathbb{E}Y_i^u Y_j^v = p_i^u p_j^v$ for any $u$, $v$. When $i = j$, $\mathbb{E}Y_i^u Y_i^u = p_i^u > (p_i^u)^2$ for $u = v$ and $\mathbb{E}Y_i^u Y_i^v = 0 < p_i^u p_i^v$. Then it is easy to see $g_{u,v}(x, y) > 1$ for $u = v$ and $g_{u,v}(x, y) < 1$ for $u \neq v$.

## D   MORE ON EXPERIMENTS

### D.1   METRICS DEFINITION

The ranking performance is evaluated through recall at K (Recall@K) and normalized discounted cumulative gain at K (NDCG@K). In our VAE-SPP model, the predicted rank of the held-out items $I_u$ for each user $u$ are obtained from sorting the intensity function $\lambda_u(x)$.

Here we keep the definition in Liang et al., (2018). Recall@K is defined as

$$\text{Recall@K} = \frac{\sum_{i=1}^{K} \mathbb{I}(r_i \in I_u)}{|I_u|}. \tag{27}$$

NDCG@K is calculated by normalizing discounted cumulative gain (DCG@K) with ideal DCG@K (IDCG@K). The definition are as follows

$$\text{DCG@K} \quad = \sum_{i=1}^{K} \frac{2^{\mathbb{I}(r_i \in I_u)} - 1}{\log_2(i+1)}, \quad \text{NDCG@K} \quad = \frac{\text{DCG@K}}{\text{IDCG@K}}, \tag{28}$$

where $\mathbb{I}$ is the indicator function and $r_i$ is the $i^{\text{th}}$ item among held-out items; IDCG@K is the ideal DCG@K when the ranked list is perfectly ranked.

### D.2   HYPERPARAMETERS

We implement our models in Tensorflow based on VAE-CF[2]. We keep the same MLP architectures and hyperparameters for both of them. We use $\beta$-VAE with as suggested. The only additional hyperparameters for our model is the $\sigma^2$ in the kernel function, which is determined using grid search on the validation set. We conducted the experiments on a single GTX 1080 TI 11GB GPU.

For simulation data, we train both models for 200 epochs using Adam optimizer with $\beta = 0.2$, $lr = 5 \times 10^{-5}$. We use mini-batches of size 20. Our architectures consist of a one layer MLP with $K = 50$. For VAE-SPP, $\sigma^2 = 0.001$. For Gowalla-NYC data, we train both models for 200 epochs using Adam optimizer with $\beta = 0.2$, $lr = 5 \times 10^{-4}$. We use mini-batches of size 20. Our architectures consist of a one layer MLP with $K = 50$. For VAE-SPP, $\sigma^2 = 1 \times 10^{-5}$. For Gowalla-LA data, we train both models for 200 epochs using Adam optimizer with $\beta = 0.2$, $lr = 1 \times 10^{-3}$. We use mini-batches of size 20. Our architectures consist of a one layer MLP with $K = 50$. For VAE-SPP, $\sigma^2 = 0.001$. For ML-1M data, we train both models for 100 epochs using Adam optimizer with $\beta = 0.2$, $lr = 1 \times 10^{-3}$. We use mini-batches of size 5. Our architectures consist of a one layer MLP with $K = 200$. For VAE-SPP, $\sigma^2 = 1 \times 10^{-5}$. For ML-100K data, we train both models for 100 epochs using Adam optimizer with $\beta = 0.2$, $lr = 1 \times 10^{-3}$. We use mini-batches of size 5. Our architectures consist of a one layer MLP with $K = 200$. For VAE-SPP, $\sigma^2 = 1 \times 10^{-5}$. The one-layer GNN in ML data is trained using GraphSAGE, for which the embedding dimension is 32 and the number of neighborhood is 10 for item and 5 for user. The graph is consist of the edges between users and items as well as the edges between items based on their Jaccard similarity.

We use python statsmodel for the KDE and GPy for TGCP. Bandwidth for KDE is selected automatically. The hyperparameters for TGCP are determined with a grid search on the validation set. For simulation data sets, we set an RBF kernel with variance=1, lengthscale=0.1 for TGCP. For Gowalla-CA data sets, we set an Matern32 kernel with variance=1e-3, lengthscale=0.1 for TGCP. For Gowalla-NYC data sets, we set an Matern32 kernel with variance=1e-4, lengthscale=0.01 for TGCP.

### D.3   ADDITIONAL EXPERIMENTS

On the training of VAE and GNN, we tried different training settings (separately or jointly) and choose to train them jointly. We also tested the point estimate version of the VAE-CF called DAE-CF (Mult-DAE in Liang et al., (2018), with the same setting), which can improve the result under

---

[2]https://github.com/dawenl/vae_cf

certain metrics. One can easily extend our work to a DAE-SPP to obtain a point estimation for the SPP intensity.

Table 6: Testing results on MovieLens-100K. These methods share the same network and trained with 100 epochs. The test data is evaluating the model with the best performance during the validation. Separate means that GNN is trained separately with VAE-SPP.

|  | NDCG@100 | Recall@20 | Recall@100 |
|---|---|---|---|
| VAE-CF | 40.88 | 32.32 | 57.63 |
| DAE-CF | 40.98 | 29.29 | 58.80 |
| VAE-SPP | 41.50 | 31.34 | 58.99 |
| VAE-SPP-Separate | 41.43 | 31.15 | 58.82 |

We also did experiments on the MLPs for VAE. For Movie Lens 1M, the larger network in VAE-CF leads to a 40.3 NDCG@100 for VAE-CF and 41.9 for VAE-SPP. As a result, we use the smaller one instead.

