# OpenReview forum: "Variational Autoencoders for Highly Multivariate Spatial Point Processes Intensities"
_ICLR.cc/2020/Conference — Accept (Poster)_

### Official Review · AnonReviewer2 · 2019-10-23
**Official Blind Review #2**

**Rating:** 6

**Review:**

In this paper, the authors propose a VAE model for spatial point processes. The model generalizes the kernel density-based intensity and applies variational inference. The model is applied to synthetic datasets, a location-based social network dataset, and a recommender system dataset.

The paper is well motivated and clearly written. I found the probabilistic modeling interesting. My major concern is that, with added complexity, the experimental results suggest that VAE-SPP is not significantly better than the existing VAE-CF on most tasks.

For the MovieLens task, it seems that the GNN is an important component in the pipeline, but no further detail is provided about it, including the network architecture. The authors might also want to provide the definitions of NDCG@k and Recall@k, at least in the appendix.

---

I appreciate the author's detailed response and updated paper. I have changed my rating.

**Experience Assessment:**

I do not know much about this area.

**Review Assessment: Checking Correctness Of Derivations And Theory:**

N/A

**Review Assessment: Checking Correctness Of Experiments:**

I assessed the sensibility of the experiments.

**Review Assessment: Thoroughness In Paper Reading:**

I made a quick assessment of this paper.

---

> ### Author Response · Authors · 2019-11-14
> **Response to Reviewer 2**
>
> We thank reviewer 2 for valuable comments. We update the paper accordingly, especially in the appendix, providing more details about the experiments and metrics.
>
> 1. “My major concern is that, with added complexity, the experimental results suggest that VAE-SPP is not significantly better than the existing VAE-CF on most tasks.”
>
> - In recommender systems, even a small percentage improvement is non-trivial, especially with the strong VAE-CF baseline here. Comparing with the state-of-the-art methods in the SPP literature, this amortized method significantly outperforms them and extends the capability of handling highly multivariate SPPs.
>
> Moreover, SPPs contain great potential for drawing inspiration from or building further on top of, for future work extending the simple intensity function considered here. Our paper establishes a foundation to build from in that regard. This has the impact of introducing a new state of the art in VAE based collaborative ﬁltering techniques that will either inspire further point process-based methods or will at the very least be used to compare against. However, we are not claiming that we solve the classic CF problem completely with the VAE-SPP. In the updated version, we modify the paper to clearly state the applicability of the current approach.
>
> 2. “For the MovieLens task, it seems that the GNN is an important component in the pipeline, but no further detail is provided about it, including the network architecture. The authors might also want to provide the definitions of NDCG@k and Recall@k, at least in the appendix.”
>
> - Thanks for the great suggestion! We are using a one-layer GNN on the graph which consists of the edges between users and items (ratings) as well as the edges between items based on their Jaccard similarity. The training of GNN is based on the GraphSAGE with a mean aggregator.
>
> More details are provided in Appendix D.2 along with all other network architectures used in this paper, including MLP in VAE and classic SPP methods. We also provide the definitions for NDCG@K and Recall@K in Appendix D.1 instead of simply citing Liang’s paper.

---

### Official Review · AnonReviewer3 · 2019-10-23
**Official Blind Review #3**

**Rating:** 8

**Review:**

The work describes an application of a spatial point process for solving problems with missing data. The authors introduce a novel method based on a non-parametric definition of point intensities for the multivariate case. The method incorporates VAE framework to effectively handle missing points via smooth intensity estimation and enjoys amortized inference for efficient computations and quick prediction generation. Using a sequence of mild assumptions, the authors show connection to a popular VAE-based collaborative filtering model, which turns out to be a special case of their approach.


This is a rigorous study providing theoretically justified evidence on the effectiveness of the proposed approach. Apart from the issues in the last part of experiments with classical collaborative filtering task (which will be detailed below), the work presents a solid research. I would therefore vote for accepting it.


The text is well structured, and all key points are clearly explained. The problem solved by the authors is well described, and the motivation for this work is convincing. The way point process theory is applied constitutes a rigorous probabilistic approach. The authors convincingly justify the need for all approximations and simplifications made in the model. One of key results making the entire model feasible is supported by the corresponding theorem proved by the authors. I haven’t carefully verified all the derivations, though.


My major concern is related to the last part with experiments on the Movielens data. As the authors state, “applications without explicit spatial information, we embed each event into a latent space as a vector.” “No spatial information” is exactly the case with the standard collaborative filtering task, which the authors attempt to solve. This leads to an introduction of an additional model like GNN, which is unrelated to the main approach. As GNN is involved it’s not immediately obvious that the improvement over standard VAE architecture, observed in the experiments on ML-100K and ML-1M, is due to a better point process modelling.  No evidence is provided to argue that this is not simply due to a good compression or a good data preprocessing achieved by a GNN architecture itself. Therefore, the results on a pure recommender systems part are not convincing. What would happen if GNN was trained and fed into another (simpler) algorithm? Maybe a simple KNN based algorithm would produce comparable or even better results? As indicated by the work of [Dacrema, Cremonesi, Jannach 2019] on “A Worrying Analysis of Recent Neural Recommendation Approaches”, VAE-CF (along with several other recently proposed neural network-based methods) is inferior to even properly tuned kNN-based models. I would not be surprised, if a kNN model trained on GNN output would produce even better results than the proposed VAE-SPP.

Another related question is how incorporating GNN affects the training time? Is it comparable to that of VAE-CF or is it much worse? Computational performance is an important part in making practical decisions and should be also considered.

Furthermore, both ML datasets used for tests are too small and not very representative to make any generalized conclusions. Even on a larger ML-20M dataset an optimal SVD-based model can be trained within several minutes on a standard CPU on a laptop (according to my experiments, VAE-CF would take at least twice longer on Tesla K80). Therefore, it can hardly be considered a realistic example. In practice, there could be millions and hundreds of millions of items. The authors even mention it in the in the introduction, using it as a vehicle to motivate their approach. However, computing similarities between that many entities can be a laborious task on its own, which adds an extra layer of complexity and again is not directly related to the main approach. It can easily become a bottleneck or make further computations inefficient. More efficient similarity computations may in contrast reduce the resulting accuracy.
The issue can get even worse, because, unlike classical MF methods, there’s still no proper support for sparse operations in NN frameworks. In the VAE framework it means that, during the training, user batches will be converted into dense arrays and may become inefficient to work with in terms of memory and CPU utilization (a few non-zero entries vs. hundreds of millions of explicitly stored zeroes).
In spite of all this, I’d also suggest rephrasing “We validate these beneﬁts through extensive experiments” as it sounds a bit exaggerated (if we are considering real recommender systems applications). I agree that the proposed approach is potentially applicable in real cases for recommender systems, however, there’s still not enough evidence for this. In fact, I don’t even think that completely removing the part with ML-100K and ML-1M datasets would make the whole work any worse. Clearly stating the region of applicability of the proposed approach would be enough. Right now some statements in this section in contrast are raising concerns rather than convincing the reader. The wording should be at least changed, so that readers do not get an impression that the case with classical CF task is solved purely by the proposed VAE-SPP approach.

Other remarks to help improve the text:
1) “… points are more likely to … form clusters than the simple Poisson process …” the sentence seems to be inconsistent.
2) “The generative process of our model can be described as follow:” -> … as follows:
3) Page 4, last paragraph, line 6 – shouldn’t the upper bound for summation be N_u instead of just N?

References:
Dacrema, Maurizio Ferrari, Paolo Cremonesi, and Dietmar Jannach. "Are we really making much progress? A worrying analysis of recent neural recommendation approaches." In Proceedings of the 13th ACM Conference on Recommender Systems, pp. 101-109. ACM, 2019.

**Experience Assessment:**

I do not know much about this area.

**Review Assessment: Checking Correctness Of Derivations And Theory:**

I assessed the sensibility of the derivations and theory.

**Review Assessment: Checking Correctness Of Experiments:**

I carefully checked the experiments.

**Review Assessment: Thoroughness In Paper Reading:**

I read the paper at least twice and used my best judgement in assessing the paper.

---

> ### Author Response · Authors · 2019-11-14
> **Response to Reviewer 3**
>
> We thank reviewer 3 for valuable comments and suggestions. The wording in the introduction, conclusion and experiment sections are changed to specify the applicability of our approach and avoid the impression that we claim to solve the CF task. Sorry about the confusion.
>
> We aim to establish a theoretical foundation for various spatial point processes models for the usage of collaborative ﬁltering. This will be signiﬁcant for researchers who wish to further this direct line of work.
>
> The interesting comparison paper of [Dacrema, Cremonesi, Jannach 2019] and many of your comments bring to us several important future works. We cite the paper and summarize our future plan in the conclusion, including investigating the performance of GNN with simpler methods and improve the scalability of the joint training approach. We address the computational cost of adding GNN in the updated paper. Training GNN separately can reduce the cost but it could harm the performance in CF applications (see Appendix D. 3). A possible solution is to perform a graph-cut in order to reduce the complexity of GNN. Finally, we fix a few typos in the text in the updated version.

---

### Official Review · AnonReviewer4 · 2019-10-31
**Official Blind Review #4**

**Rating:** 6

**Review:**

ICLR review

In this paper, the authors propose to tackle the multivariate spatial point process model with a variational inference approach. The variational inference is implemented with MLP (amortized inference). In experiments, the results show that the proposed approach outperforms VAE-based collaborative filtering on Gowalla datasets and MovieLens datasets.

My overall judgement of this paper leans to acceptation. However, by doing a quick comparison with Liang's VAE-CF paper. The proposed algorithm seems to be an extension or modification of Liang's framework. Therefore I guess the main contribution of this paper is Eq. (9). If so, I'm expecting more detailed comparison with VAE-CF in the paper (not only quantitive evaluation).

Questions:

- In Liang's paper, I see they report the evaluation results on both ML-1M and ML-20M dataset. However, in this paper, the results are reported on ML-100K and ML-1M. Why not evaluate the model on the bigger ML-20M dataset?

- In Liang's paper and also He's paper (Neural Collaborative Filtering), they report NDCG@10 on ML-1M dataset. The NCF approach reaches a NDCG@10 of 0.426, Multi-DAE (denoising version of Liang's model) reaches 0.446. You are reporting NDCG@100, which is a different measure. However, the numbers in Table 4 seem to be lower than what I expected. Can you compare with them in the same condition?

**Experience Assessment:**

I do not know much about this area.

**Review Assessment: Checking Correctness Of Derivations And Theory:**

I did not assess the derivations or theory.

**Review Assessment: Checking Correctness Of Experiments:**

I carefully checked the experiments.

**Review Assessment: Thoroughness In Paper Reading:**

I read the paper thoroughly.

---

> ### Author Response · Authors · 2019-11-14
> **Response to Reviewer 4**
>
> We thank reviewer 4 for valuable comments. The paper is updated to address the questions, including additional experiments and a paragraph discussing the relationship between our model and VAE-CF after Thm. 1. As you mentioned here, our VAE framework incorporating SPPs as the underlying generating mechanism is a generalization of VAE-CF, which is, in fact, a special case when one ignores the spatial heterogeneity (use a delta function). This is one of the main contributions of our work. Moreover, the SPP likelihood in Eq. (9) cannot be directly applied to VAE due to the definition of the hidden variable. Another main contribution of this paper is to develop the alternative model that uses VAE for the amortized inference of spatial point process and show the equivalence of VAE-CF and VAE-SPP with Thm. 1.
>
> 1. “In Liang's paper, I see they report the evaluation results on both ML-1M and ML-20M dataset. However, in this paper, the results are reported on ML-100K and ML-1M. Why not evaluate the model on the bigger ML-20M dataset?”
>
> - Thanks for mentioning this. We will work on this in the future, especially on the joint training of GNN and VAE-SPP for the larger datasets, as mentioned in the updated paper. The main focus of this paper is on highly multivariate SPPs and it is showing potentials on collaborative filtering. We modify several sections of the paper to clarify that we are not claiming that the collaborative filtering problem is completely solved here. Instead, we hope to inspire further point process-based CF methods based on this framework.
>
> One way to reduce the computational complexity is to obtain the embedding separately.  However, based on our experiments, if the embeddings of items are obtained separately, then the gap between the VAE-CF and VAE-SPP models becomes smaller. For Movie Lens 100k, training GNN and VAE separately leads to a smaller NDCG@100 (see Appendix D). We would love to explore other approaches such as graph-cut based methods to reduce the computational complexity of GNN.
>
> 2. “In Liang's paper and also He's paper (Neural Collaborative Filtering), they report NDCG@10 on ML-1M dataset. The NCF approach reaches a NDCG@10 of 0.426, Multi-DAE (denoising version of Liang's model) reaches 0.446. You are reporting NDCG@100, which is a different measure. However, the numbers in Table 4 seem to be lower than what I expected. Can you compare with them in the same condition?”
>
> - In fact, there are two different ways of training in VAE-CF and NCF paper. In NCF, they split the data in (user, item) level to learn the embedding for all users in the data. In VAE-CF, the training-validation-test splitting is done at the user level, i.e. users are split into train/test/validation sets: Users with their full rating histories are used to train the model; For testing and validation users, the model is trying to infer the held-out ratings based on part of their history. In the area of SPPs, the setting is more similar to the case of VAE-CF where one wants to infer the missing data for subprocesses that are unseen during the training. As a result, our experiments in ML are *not* comparable to results using the NCF setting such as Table 3 in Liang’s paper (see section 4.4 in Liang’s paper about their splitting methods when comparing with NCF). The choice of different splitting methods definitely plays an important role in the NDCG numbers and is worth further investigation, which is however not the focus of this paper.
>
> We add the additional experiments comparing with DAE-CF (Multi-DAE), a point estimation version of VAE-CF, in the setting of user splitting. In terms of the NDCG@100, the DAE-CF is 0.4098 for ML-100K (see Appendix D for more details) and 0.4153 for ML-1M, which is better than VAE-CF but lower than VAE-SPP. Moreover, we believe that a better comparison with the DAE-CF could be a point estimation version of VAE-SPP (DAE-SPP?) for CF applications. It could achieve an even better NDCG number in these experiments as the data is less sparse.

---

### Decision · Program_Chairs · 2019-12-19

**Decision:**

Accept (Poster)

**Comment:**

This paper presents a novel VAE-based model for multivariate spatial point process which can realize efficient inference by amortization and handle missing points via smooth intensity estimation. Authors also provide interesting theoretical analysis to connect their method to a popular VAE-based collaborative filtering method.
Overall, all reviewers appreciate the methodological and theoretical contributions of the paper. During the reviewer discussion, one reviewer decided to update to the score to Weak Acceptance. There are still some concerns regarding experimental validation, I think the paper provides enough theoretical contribution to the community and would like to recommend acceptance.